# Quantifying the inhibitory impact of soluble phenolics on anaerobic carbon mineralization in a thawing permafrost peatland

**Alexandra B. Cory**[1]*, **Jeffrey P. Chanton**[1], **Robert G. M. Spencer**[1], **Olivia C. Ogles**[1], **Virginia I. Rich**[2], **Carmody K. McCalley**[3], **IsoGenie Project Coordinators**[¶], **EMERGE 2021 Field Team**[¶], **Rachel M. Wilson**[1]

1 Department of Earth, Ocean, and Atmospheric Science, Florida State University, Tallahassee, FL, United States of America, 2 Department of Microbiology, The Ohio State University, Columbus, OH, United States of America, 3 Thomas H. Gosnell School of Life Sciences, Rochester Institute of Technology, Rochester, NY, United States of America

¶ Membership of the IsoGenie Project Coordinators is listed in the Acknowledgments.
¶ Membership of the EMERGE 2021 Field Team is listed in the Acknowledgments.
* alexandra.cory.b@gmail.com

**Data Availability Statement:** All relevant data are within the manuscript and its Supporting Information files.

## Abstract

The mechanisms controlling the extraordinarily slow carbon (C) mineralization rates characteristic of *Sphagnum*-rich peatlands ("bogs") are not fully understood, despite decades of research on this topic. Soluble phenolic compounds have been invoked as potentially significant contributors to bog peat recalcitrance due to their affinity to slow microbial metabolism and cell growth. Despite this potentially significant role, the effects of soluble phenolic compounds on bog peat C mineralization remain unclear. We analyzed this effect by manipulating the concentration of free soluble phenolics in anaerobic bog and fen peat incubations using water-soluble polyvinylpyrrolidone ("PVP"), a compound that binds with and inactivates phenolics, preventing phenolic-enzyme interactions. $CO_2$ and $CH_4$ production rates (end-products of anaerobic C mineralization) generally correlated positively with PVP concentration following Michaelis-Menten (M.M.) saturation functions. Using M.M. parameters, we estimated that the extent to which phenolics inhibit anaerobic $CO_2$ production was significantly higher in the bog—62 ± 16%—than the fen—14 ± 4%. This difference was found to be more substantial with regards to methane production—wherein phenolic inhibition for the bog was estimated at 54 ± 19%, while the fen demonstrated no apparent inhibition. Consistent with this habitat difference, we observed significantly higher soluble phenolic content in bog vs. fen pore-water. Together, these findings suggest that soluble phenolics could contribute to bogs' extraordinary recalcitrance and high (relative to other peatland habitats) $CO_2$:$CH_4$ production ratios.

## Introduction

Due to the enormous quantity of carbon (C) contained in peatlands—current estimates ranging from ~530–1,175 Pg globally (equivalent to ~60% - 134% of current atmospheric C stores)

**Funding:** Grant 1 Recipient: V.I.R. Grant number: DE-SC0010580 Funding Source: the Genomic Science Program of the United States Department of Energy Office of Biological and Environmental Research Grants URL: https://genomicscience. energy.gov/ Grant 2 Recipient: V.I.R. Grant number: DE-SC0016440 Funding Source: the Genomic Science Program of the United States Department of Energy Office of Biological and Environmental Research Grants URL: https:// genomicscience.energy.gov/ Grant 3 Recipient: V.I. R. Grant Number: 2022070 Funding Source: the EMERGE Biology Integration Institute of the National Science Foundation URL: https://www.nsf. gov/funding/pgm_summ.jsp?pims_id=505684 The funders had no role in study design, data collection and analysis, decision to publish, or preparation of the manuscript.

**Competing interests:** The authors have declared that no competing interests exist.

[1–3]—shifts in peatland C cycling have potentially significant impacts on the global climate. Most peatlands are $CO_2$ sinks and $CH_4$ sources [4–6]. The former process is cooling to the climate, while the latter has a warming effect. While peatland C deposition has had a significant cooling effect on the climate through much of the Holocene, this effect has diminished over the last ~150 years. Estimates of present climatic impacts range from slightly cooling (-0.7 W x $m^{-2}$; instantaneous box-model estimate) [7] to slightly warming (+0.6 Pg $CO_2$-equiv $y^{-1}$; field flux estimate) [5].

This regime shift is the result of climate change-induced disruptions to the peatland system including water table shifts, permafrost thaw, and increased frequency of fire and drought [8–10]. Acute anthropogenic disturbances (e.g., drainage and burning) have also created significant C balance disruptions and will likely continue to do so without political intervention [5]. Together, these disruptions may speed up rates of $CO_2$ and $CH_4$ production via decomposition, thereby shifting peatlands into significant drivers of warming. It is thus imperative that we incorporate an accurate assessment of peatland-climate dynamics into global climate models. To do so, we must understand the underlying biogeochemical processes responsible for peat C mineralization.

Due to the extent of water saturation typically observed in peatlands, most of the peat column decomposes anaerobically [11]. The resulting anoxia precludes oxic respiration and the generally nutrient-depleted conditions characteristic of peatlands hinder respiration via inorganic terminal electron acceptors ("TEAs") [11]. As such, anaerobic decomposition is limited primarily to three low energy-yielding processes (1) hydrolysis (breakdown of complex organic compounds into simple compounds), (2) fermentation, and (3) methanogenesis. Though slow, anaerobic decay has the potential to significantly impact the climate due to the production of methane (which has a global warming potential of 45 times that of $CO_2$ on a 100-year timescale [12]. This study will, therefore, focus on the mechanisms controlling anaerobic decomposition in peatlands which, for the sake of brevity, will be henceforth referred to as "decomposition" or "C mineralization".

Peatland permafrost thaw results in a mosaic of habitat types with differing hydrological and pH regimes that have significant effects on decomposition rates [13–17]. Of particular significance is the shift from *Sphagnum*-rich ("bog") to sedge-dominated ("fen") habitats—the former being known for extraordinarily slow decomposition rates and high $CO_2$:$CH_4$ production ratios—and the latter characterized by relatively quicker decomposition and lower $CO_2$: $CH_4$ ratios [16, 18]. Certain abiotic factors partially explain the slower decomposition rates typically observed in bogs, such as lower pH (~4.5 in bogs, 7–8 in fens; [19]) and availability of TEAs. However, they do not fully account for the much slower decomposition rates observed in bogs [11, 20, 21].

Soluble phenolics have been invoked as potentially significant inhibitors of bog decomposition due to (1) the abundance of bog dissolved organic matter ("DOM") with high (relative to fen) aromaticity, molecular weight, and O/C ratios—indicative of high soluble phenolic content [22] and (2) the propensity of soluble phenolics to suppress microbial metabolism and inhibit cell growth [22–27]. Metabolic disruptions are attributed to phenolic-enzyme interactions (bonding and/or adsorption), which limit enzyme activity [28–33]. Disruptions to cell growth and function are attributed to phenolic-membrane interactions, which can cause membrane injury and increased permeability [34]. The latter is associated with increased influx of extracellular compounds—some of which can be toxic to micro-organisms—and increased efflux of intracellular components that are necessary for cell growth, such as proteins, potassium, and phosphates [34–38].

Though the inhibitory effects of soluble phenolics are generally accepted, the extent to which they inhibit C mineralization in bogs remains unclear [9, 24, 25, 39, 40]. Studies to date

have focused heavily on the potential for soluble phenolics to inhibit enzymatic hydrolysis. These studies have yielded conflicting results, ranging from inconsequential [39, 40] to significant inhibition of hydrolysis [9, 24, 25]. To clarify the impacts of soluble phenolics on bog C mineralization, it is necessary to consider all three stages of bog decomposition (hydrolysis, fermentation, and methanogenesis). There are three reasons for this need: (1) inhibition of fermentation and methanogenesis by soluble phenolics has been observed [27, 35, 41, 42]; (2) evidence of simple sugar buildup in bog peat indicates that C mineralization rates are sometimes not limited by hydrolysis (which produces simple sugars), but rather fermentation and/or methanogenesis (which collectively consume simple sugars) [43]; and (3) there is not always a significant relationship between respiration rates and hydrolytic enzyme activities [44].

Studies that consider the impacts of soluble phenolics on all three stages of bog decomposition are scant. Suppression of $CO_2$ and $CH_4$ production—end-products to all three stages— has been observed in incubated peat amended with phenolic-rich DOM [45]. Though it is feasible that soluble phenolics caused this inhibition, the presence of other potentially inhibitory DOM compounds precludes definitive affirmation of this effect [45]. Suppression of aerobic respiration by soluble phenolics has been observed in an aerobic incubation experiment [9], but only after the addition of respiration substrates (C and nutrients). This finding suggests that the potentially inhibitory impacts of soluble phenolics are inconsequential in substrate-limited settings. Given that substrate supply varies in response to site-specific factors—such as vegetation and climate—it is necessary to broaden assessments regarding the impact of soluble phenolics on C mineralization to more sites. Moreover, given the climatically significant role of anaerobic decomposition, [11], it is necessary to apply these assessments to anaerobic conditions.

We analyzed the relationship between *in situ* soluble phenolic content and anaerobic C mineralization rates in a bog and fen site within a Swedish permafrost peatland (Stordalen Mire). We used the cumulative production of $CO_2$ and $CH_4$ (the end-products of C mineralization in anaerobic bog environments) to determine C mineralization rates. Using the methods of [46], we manipulated the concentration of free soluble phenolics using water-soluble polyvinylpyrrolidone ("PVP"). This synthetic polymer "inactivates" soluble phenolics via binding and precipitation, preventing phenolic-enzyme interactions from occurring [47]. To maximize this inactivation, we sought to saturate our incubations with PVP. As the concentration necessary to achieve saturation in our incubations was unknown, additions were undertaken across a wide concentration range.

We hypothesized that increasing PVP concentration would increase C mineralization rates in the bog without the addition of respiration substrates because substantial buildup of simple sugars has been observed in bog pore-water from our study site [43]. We hypothesized that the bog would contain higher soluble phenolic content than the fen, given prior observations that bogs possess higher O/C ratios, greater aromaticity, and higher molecular weights—suggesting higher (potentially inhibitory) phenolic content [22]. We hypothesized that higher soluble phenolic content would cause a stronger inhibitory effect on bog vs. fen C mineralization.

We hypothesized that the relationship between PVP concentration and C mineralization rates would follow a Michaelis-Menten saturation function (Fig 1). By examining the effects of PVP saturation on C mineralization, we sought to quantify the extent to which soluble phenolics inhibit bog and fen C mineralization. We expect these to be minimum estimates given that (1) even in PVP-saturated conditions, a minute portion of free soluble phenolics likely persist [47] and (2) of these persisting soluble phenolics, some could feasibly continue interacting with enzymes, leading to continued inhibition of C mineralization.

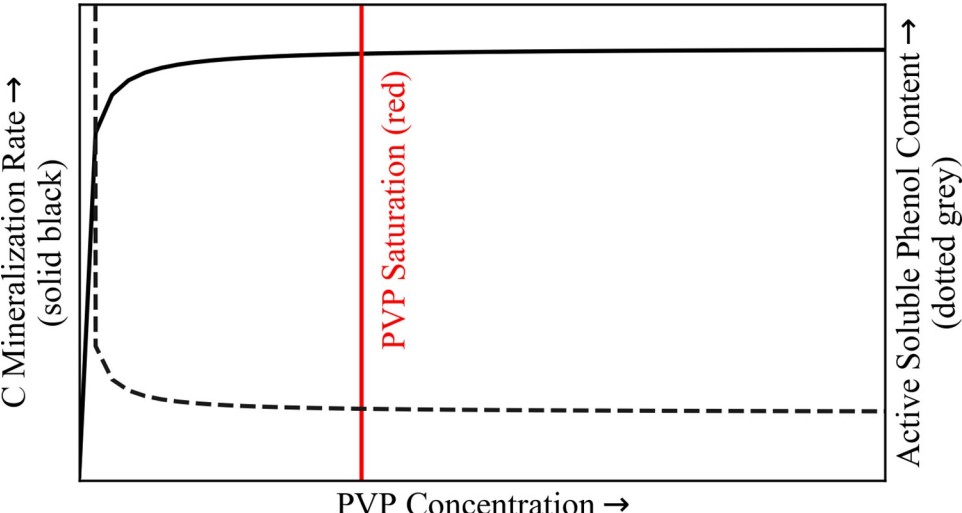

**Fig 1. Hypothesized relationship between polyvinylpyrrolidone (PVP) concentration vs. C mineralization.** C mineralization rate (measured by $CO_2$ and $CH_4$ production) corresponds to the primary y axis (solid black line). Assumed active soluble phenol content corresponds to the secondary y axis (dotted grey line). Addition of PVP was hypothesized to positively impact $CO_2$ and $CH_4$ production rates by inactivating soluble phenolics which would otherwise inhibit C decomposition. This relationship was expected to follow a Michaelis-Menten saturation curve. After reaching a point of PVP saturation (red line), further increases in PVP concentration were expected to yield no significant changes in C mineralization rates.

## Site description

Stordalen Mire is a subarctic peatland approximately 10 km east of Abisko, Sweden (68.35˚, 19.05˚E). Located within the discontinuous permafrost zone, it is comprised of three dominant habitats—palsas, bogs, and fens—reflecting various stages of thaw with different hydrologic regimes. For an in-depth description of these habitats, see [22, 48, 49].

Palsas are characterized by raised topography situated above permafrost peat. This creates dry, ombrotrophic conditions. Vegetation is dominated by Ericaceous and woody shrubs, *Eriophorum vaginatum*, and lichens. Decay rates in these environments are slower than photosynthetic uptake, constituting a C sink [48]. Methane emissions are generally lowest in this habitat as their dry conditions limit anoxic decomposition [50].

Bogs are considered the product of slow thawing of the upper permafrost layer. They are perched above the regional water table separated by a layer of permafrost. As such, water inputs are limited to rainfall ("ombrotrophic"). Vegetation is dominated by mosses (mainly *Sphagnum* spp.), with some sedges (e.g. *Eriophorum vaginatum*, *Carex rotundata*). The pH in bogs typically ranges from 4–5, with an average value of 4.1 in Stordalen [19]. Decay rates are significantly lower than photosynthetic uptake, resulting in a significant C sink [16, 48]. While bogs are net $CH_4$ sources, their $CO_2$:$CH_4$ production ratios are $>> 1$ [18, 51] which limits their net radiative forcing [16, 18, 50, 51].

Fens are devoid of intact permafrost. Land subsidence causes the water table to rise above the soil surface allowing for lake-water and/or groundwater input. Vegetation is dominated by sedges such as *E. angustifolium* and *Carex rostrata*. *Sphagnum* spp. is less abundant than in bogs. Fens at Stordalen are distinct from previously studied subarctic mires in that they contain more *Sphagnum* spp. and have lower pH values (~5.1 at Stordalen, 7–8 for previously studied sites; [19]). Decay rates are faster than in bogs but still slower than photosynthetic C uptake, constituting a weak C sink [48]. This cooling effect is typically offset by the relative

high production of methane ($CO_2$:$CH_4$ ratios generally closer to 1), resulting in higher warming potential relative to bogs [16, 50, 52].

## Methods

Under the IsoGenie Project, our team maintains field permits with the Abisko Scientific Research Station and its parent organization, the Swedish Polar Research Secretariat, to collect samples from Stordalen Mire.

### Field collection

Peat for the laboratory incubation experiments was obtained from two sites: one bog (19.04923˚N, 68.35559˚W) and one fen (19.04620˚N, 68.35443˚W). These sites were selected because they were (1) determined to be representative of the plant community and hydrologic conditions of their respective habitats and (2) within 50 m of one another. Each core was collected in July 2018 using an Eijelkamp perforated stainless steel corer (Eijelkamp, The Netherlands) as described in [53]. Specific core locations were selected at random, and attention was paid carefully not to sample in previously cored sites. The lengths of the bog and fen cores were 27 and 24 cm, respectively, with whole-core diameters of 5 cm. Each core was sectioned using a razor blade and a subsection from 9–19 cm depth was set aside for incubation analysis. This depth range was selected because at this depth peat is generally below the water table, facilitating anaerobic decomposition (which is the focus of this study). The 9–19 cm sub-section weights for bog and fen were 933.7 and 651.2 g, respectively. The peat was stored at -20˚C from the point of collection up until the experiment start date—20 and 29 months for bog and fen, respectively.

Pore-water for soluble phenolic analysis was obtained from four sites within Stordalen Mire: two bog (site 1: 19.04758˚N, 68.35330˚W, site 2: 19.04923˚N, 68.35559˚W) and two fen (site 1: 19.04658˚N, 68.35337˚W, site 2: 19.04620˚N, 68.35443˚W). Pore-water was obtained by pushing stainless steel piezometers to the desired depth interval and extracting the water with an airtight syringe. At each site, three pore-water profiles were obtained, each of which included extractions from up to four depth intervals (depending on water table levels and compaction of peat, which can prevent retrieval through manual suction when too high). These depth intervals were: 1–5, 10–14, 20–24, 30–34, and 40–44 cm below ground surface. After extraction, the samples were filtered to 0.7-μm and stored at -20˚C until analysis.

### Experimental design

We prepared replicate incubation experiments on bog and fen peat. For each habitat, we prepared six treatments. Each had three replicates, totaling 18 vials. Treatment 1 was an untreated control. Treatments 2–6 spanned a polyvinylpyrrolidone (PVP) concentration range of 0.001–0.064 g/mL (Table 1), which extended above and below concentrations shown to stimulate enzymatic hydrolysis in DOM-rich arctic river water (0.005 and 0.010 g/mL) [46].

**Table 1. PVP Concentration by treatment number.**

| Treatment # | PVP Conc. (g/mL) |
|---|---|
| 1 | 0 |
| 2 | 0.001 |
| 3 | 0.004 |
| 4 | 0.016 |
| 5 | 0.032 |
| 6 | 0.064 |

We thawed and homogenized the peat by de-clumping it with gloved hands and forceps using the method developed by [54]. We aliquoted 40 g of homogenized peat and 30 mL deionized (DI) water into each 160-mL clear borosilicate vial. We then capped the vials with rubber septa and sealed them with aluminum crimps. To create anoxic conditions, we vigorously shook the vials and flushed the headspace with $N_2$ gas. We repeated this process until headspace $CO_2$ concentrations measured less than 0.1% (see "Gas Analysis"), which was two magnitudes lower than the $CO_2$ production measured during the experiment. The above-gauge headspace pressure was ~3.5 psi immediately following headspace flushing. We stored the vials in total darkness at room temperature (20–22˚C) and allowed them to sit for a 25-day pre-incubation phase. The purpose of this step was two-fold. First, it has been shown to facilitate consumption of molecular $O_2$ and reduction of oxidized chemical species [48, 49]. Second, it offers a re-acclimation period for the microbial community [55]. We reference all time points relative to the end of this pre-incubation period such that day 1 corresponds to the first day following the 25-day pre-incubation phase.

Two experiments were conducted—one bog, one fen—using shallow (9–19 cm) peat that was held in dark, anoxic conditions. For each experiment, six treatments were applied, with respective concentrations included above. Each treatment was conducted in triplicate, totaling 18 vials per experiment.

On day 1, water-soluble polyvinylpyrrolidone PVP was added (Sigma Aldrich, CAS #: 9003-39-8, average molecular weight: 40,000). Using a stock solution of PVP at 0.256 g/mL in DI water, we performed serial dilutions to obtain the final PVP concentrations detailed in Table 1, except for the control (treatment 1), which was composed entirely of DI water. We injected 10 mL of the relevant PVP solution (except for Treatment 1, which contained only DI water) into each vial to achieve the final concentrations indicated in Table 1. The final ratio of grams wet peat: mL solution was 1:1, which was enough to fully saturate the peat. The pH at the beginning and end of the experiment was 4.5 and 5.5 for bog and fen peat, respectively. As these values were consistent with the field, pH alteration was not necessary.

After addition of the PVP solution, we re-flushed the headspace with $N_2$ gas and shook the vials until we once again measured headspace $CO_2$ concentrations less than 0.1%. At this time, the above-gauge headspace pressure was once again ~3.5 psi. We periodically measured headspace pressure to ensure that it did not fall below 0.5 psi (to prevent air infiltration). Since the volume extracted for gas analysis ($\leq$ 250 µl) was a small fraction of the total headspace volume (80 mL), gas replacement was not necessary to maintain >0.5 psi headspace pressures. We collected headspace samples for analysis of $CO_2$ and $CH_4$ concentrations every 3–10 days throughout the duration of the experiment (56 days; consistent with [16]). At the end of the experiment, we removed the vial caps and dried the samples at a constant 68˚C. Once the sample weights stabilized, we obtained final dry weights, which we used to calculate $CO_2$ and $CH_4$ production per g dry peat (see "Statistical Analysis").

## Gas analysis

We performed all $CO_2$ and $CH_4$ concentration analyses via Flame-Ionization-Detector Gas Chromatography (GC-FID) equipped with a methanizer using methods established by [22]. We used a gas-tight syringe for injection of all samples and standards. The GC flow rate was 30 mL/min, and the temperatures were 140, 160, and 380˚C for the column, detector, and methanizer, respectively. On each sampling day, we created a linear calibration curve for both $CO_2$ and $CH_4$ using standards of known concentrations. Before sample analyses, we shook the vials vigorously to liberate gases trapped in the peat pore-spaces. We also recorded headspace pressures to calculate partial pressures of $CO_2$ and $CH_4$ (which were essential for statistical analysis).

## Soluble phenolic analysis

We used the Folin-Ciolcateu (FC) method to quantify total soluble phenolic abundance, as described by [56]. Briefly, we added 0.15 mL of pore-water to 0.5 mL of 0.27M FC reagent (diluted from starting concentration of 2M, obtained from VWR, cat# IC19518690). We mixed this solution and allowed it to sit for 5 minutes before adding 0.5 mL of 93.6 g/L $Na_2CO_3$ solution (VWR, cat# 97061–296). The samples were once again mixed and allowed to sit for 2–3 hours. Their absorbance was then measured at 765 nm using a UV/Vis absorption spectrophotometer (BioTek Synergy HTX Multi-Mode Reader). We used a standard calibration curve of gallic acid using seven concentrations ranging from 0.27 to 6.6 g/mL. Absorbance values were all blank-corrected and concentrations were adjusted based on the dilution imposed by reagent addition. The detection limit for this methodology was 2.5 μM.

## Statistical analysis

**Soluble phenolic content.** To estimate the impact of three factors upon soluble phenolic abundance—(1) habitat, (2) depth, and (3) site—we performed a 3-factor ordinary least-squares regression analysis using the ols function within the formula.api package from Python's statsmodel library. This function had two inputs: (1) formula, and (2) dataset. The formula was as follows: soluble phenolic content = a*habitat + b*depth + c*site, where a, b, and c represent coefficients determined by the ols function. (The dataset for the bog and fen are included in the S1 File).

**Gas production rates.** We calculated average production rates (μmoles × g dry peat$^{-1}$ × day$^{-1}$) for $CO_2$ and $CH_4$ using the steps outlined below. We determined total C mineralization rates ("$C_{tot}$") by taking the sum of $CO_2$ and $CH_4$ production rates.

To calculate gas production rates, we first determined the quantity of gas injected into the GC—$n_{gas(inj)}$—by inputting sample peak amplitudes into our standard calibration curve (Eq 1). We used the ideal gas law to determine the total moles injected—$n_{tot(inj)}$ (Eq 2). We calculated the gas fraction—$F_{gas}$—using Eq 3, which we used to calculate headspace partial pressures—$P_{gas}$ (Eq 4). We then applied this value to the ideal gas law to quantify the headspace moles—$n_{gas(HS)}$ (Eq 5). Using Henry's Law (Eq 6), we calculated the dissolved gas concentration—$C_{gas(aq.)}$—which we used to determine the moles in the aqueous phase—$n_{gas(aq)}$ (Eq 7). For $CO_2$ the higher pH in the fen resulted in an appreciable quantity of the dissolved inorganic C as bicarbonate requiring us to calculate the total DIC from equilibria equations. Once the total moles of $CO_2$ and $CH_4$ in the aqueous phase were calculated, we determined the moles per vial—$n_{gas}$ × vial$^{-1}$—using Eq 8. Our final daily production values—$n_{gas}$ × g$^{-1}$—were obtained by Eq 9 (where g = dry peat weight).

$$(n_{gas(inj)} = (m \cdot amplitude_{gas}) + b \tag{1}$$

$$n_{tot(inj)} = \frac{P \cdot V}{R \cdot T} \tag{2}$$

$$F_{gas} = \frac{moles_{gas-injected}}{moles_{total-inject}} \tag{3}$$

$$P_{gas} = F_{gas} \cdot P_{tot} \tag{4}$$

$$n_{gas(HS)} = \frac{(P_{gas} \cdot V_{HS})}{(R \cdot T_{vial})} \tag{5}$$

$$C_{gas(aq.)} = K_H \cdot P_{gas} \tag{6}$$

$$n_{gas(aq.)} = C_{gas_{aq}} \cdot V_{aq} \tag{7}$$

$$\frac{n_{gas}}{vial} = n_{gas(aq)} \cdot n_{gas(HS)} \tag{8}$$

$$\frac{n_{gas}}{g} = \frac{n_{gas}}{vial} \cdot \left(\frac{g}{vial}\right)^{-1} \tag{9}$$

Production rate time series were best approximated using linear regression equations. We determined average production rates ($n_{gas} \times g^{-1} \times d^{-1}$) using the slopes of these equations. We calculated respective $R^2$ values using the Microsoft Excel RSQ function. To assess the significance of PVP concentration on production rates, we used the Microsoft Excel 2-tailed T-test (T.test) function.

### Modeled response to PVP addition

The relationship between gas production rate and PVP concentration was best approximated using Michaelis-Menten equations. Since production rates were expected to be nonzero in controls (where PVP = $0 \text{ g} \times \text{mL}^{-1}$) we appended a y-intercept to the Michaelis-Menten equation (resulting in Eq 10). The y-intercept—"$Prod_0$"—was equivalent to the average production rate of controls (n = 3).

$$y = \frac{(v_{max} \cdot x)}{(k_m + x)} + Prod_0 \tag{10}$$

We used the curve_fit function from the Optimize package within Python's SciPy library to determine the values of Michaelis-Menten constants $v_{max}$ and $k_m$. We determined the production rates in PVP-saturated peat—$Prod_{sat}$—by summing the y-intercept ($Prod_0$) and $v_{max}$ using Eq (11).

$$Prod_{sat} = v_{max} + Prod_0 \tag{11}$$

We used non-parametric bootstrapping (1,000 simulations) to calculate the standard deviation (95% confidence interval) for calculated $v_{max}$ and $k_m$ values. In addition to the Optimize package (referenced above), these simulations were programmed using the Numpy and Pandas Python data packages. To determine the fraction of observed variance explained by the Michaelis-Menten model, we calculated the $R^2$ value for measured vs. modeled production rates. Outliers were identified when the squared residual for measured production rates (representing one incubation vial at one PVP concentration) deviated from modeled values by more than two standard deviations. After outliers were determined, Michaelis-Menten parameters (and associated standard deviations) were re-calculated using non-outliers.

### Results

When production rates of all three of $CO_2$, $CH_4$, and $C_{tot}$ ($CO_2$+$CH_4$) follow similar trends, we will refer to them collectively as "$GHG_c$" production. $GHG_c$ production rates generally

increased with PVP concentration (with the exception of $CH_4$ production from fen peat; Fig 2). $GHG_c$ production was linear with time for all incubations ($R^2 \geq 0.810$). $CO_2$:$CH_4$ production ratios were significantly ($p < 0.001$) elevated relative to median values between days 1–10 —a phenomenon that we attributed primarily to increasing $CH_4$ production rates (discussed in [18]). After this period, $CO_2$:$CH_4$ production ratios stabilized (as did $CH_4$ production rates). We include only data collected after the stabilization period ended (day 10) in our calculations of average $CO_2$:$CH_4$ production ratios.

## Bog incubations

Amended Michaelis-Menten models relating PVP concentration to $GHG_c$ production (Eq 10, Methods) accounted for 83%, 62%, and 83% of the observed variance in bog $CO_2$, $CH_4$, and $C_{tot}$ production, respectively (95% confidence interval, Fig 2A–2C, Michaelis-Menten parameters detailed in Table 2). One outlier was identified (see Methods, Modeled Response to PVP Addition), representing the $GHG_c$ production rate from one vial at PVP = 0.064 g/mL. PVP-saturated production rates ("$Prod_{sat}$"; calculated using Eq 11, Methods) were significantly ($p < 0.001$) higher than control production rates ("$Prod_0$"; calculated by averaging control production rates), amounting to a 2.8, 2.6, and 2.8-fold increase in $CO_2$, $CH_4$, and $C_{tot}$ production, respectively. $CO_2$:$CH_4$ production ratios ranged from 2.3–4.0 across all PVP concentrations studied, with control ratios averaging 3.3 ± 0.4. We approximated the relationship between $CO_2$:$CH_4$ vs. PVP concentration using a linear regression fit and found no significant correlation ($R^2 = 0.414$, $p = 0.65$) between these factors.

## Fen incubations

Michaelis Menten saturation functions approximating PVP vs. $CO_2$ and $C_{tot}$ production rates accounted for 88% and 80% of the observed variance in fen $CO_2$ and $C_{tot}$ production, respectively (95% confidence interval, Fig 2, Table 2). One outlier was identified, representing the $GHG_c$ production rate from one vial at PVP = 0.004 g/mL. Michaelis-Menten equations could not be approximated for PVP vs. $CH_4$ production rates in fen incubation. No other function relating these two parameters was found to yield a significant relationship.

While PVP-saturated production rates for $CO_2$ and $C_{tot}$ ($Prod_{sat}$; calculated using Eq 11, Methods) were significantly ($p < 0.001$) higher than control production rates ($Prod_0$; calculating by averaging control production rates), the extent of increase from PVP saturation was significantly ($p < 0.001$) lower than in the bog, amounting to 1.2 and 1.1-fold increases following saturation, respectively. $CO_2$:$CH_4$ production ratios ranged from 2.7–4.4 across all PVP concentrations studied, with control ratios averaging 2.72 ± 0.04. We approximated the relationship between $CO_2$:$CH_4$ vs. PVP concentration using a linear regression fit and found that, though this fit explained only 28.4% of observed variance in $CO_2$:$CH_4$ ratios, there was a significant positive correlation between these factors.

## Quantifying phenolic inhibition of C mineralization

We calculated the extent to which phenolics apparently inhibited $GHG_c$ production in incubated bog and fen peat using the following equation:

$$\text{Phenolic Inhibition} = 100 \cdot \left( \frac{\text{Prod}_{sat} - \text{Prod}_0}{\text{Prod}_{sat}} \right) \tag{12}$$

The reasoning behind this equation was as follows. PVP saturation is considered a method for decreasing soluble phenolic effectiveness. As such, $GHG_c$ production in PVP-saturated peat

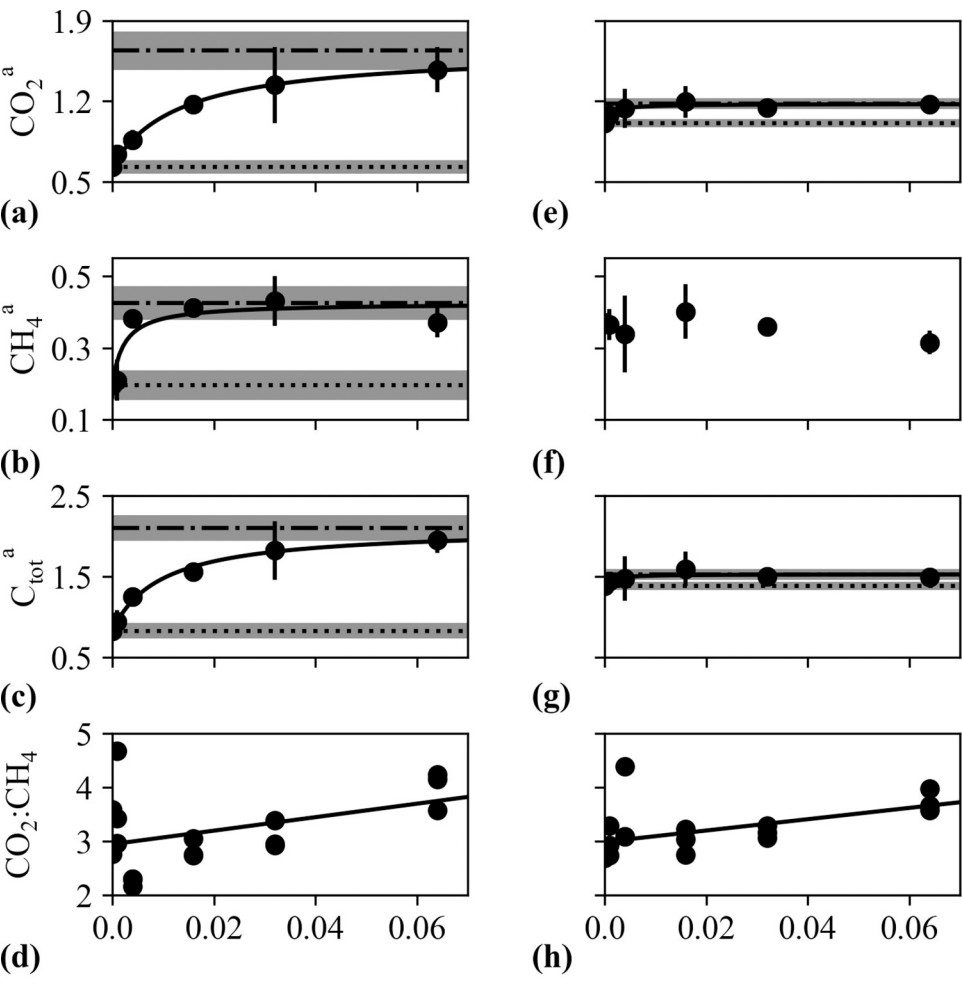

**Fig 2. Observed and modeled impacts of polyvinylpyrrolidone (PVP) on C mineralization.** PVP concentration (g mL$^{-1}$) vs. gas production in incubated bog (a-d) and fen (e-h) peat. $^a$'$CO_2$' (panels a,d), '$CH_4$' (panels b,f), and '$C_{tot}$' ($CO_2+CH_4$; panels c,g) refer to production rates (μmoles× g dry weight$^{-1}$ × d$^{-1}$) while '$CO_2:CH_4$' (panels d and h) are unitless ratios. Measured and modeled values are displayed as filled circles and solid lines, respectively. Modeled values were calculated using an amended Michaelis-Menten function (Eq 10, Methods) for panels a-c and e-g ($R^2$ values provided in Table 2). Modeled values were calculated using best-fit linear regression curve for panels d ($R^2 = 0.414$ p = 0.65) and h ($R^2 = 0.284$ p<0.001). "$Prod_{sat}$"—the estimated production rate for PVP-saturated peat (Eq 11, Methods)—and "$Prod_0$"—the average control production rate—are displayed as dashed and dotted lines, respectively (panels a-c and e-g). Standard deviations for $Prod_0$ and $Prod_{sat}$ are depicted with gray shading.

($Prod_{sat}$) should theoretically approximate peat that is free of active soluble phenolics, except for the minute portion that persists in solution following equilibration with PVP. The %Phenolic Inhibition by phenolics should thus be measured by a comparison between PVP-saturated ($Prod_{sat}$) and control ($Prod_0$) peat. As it is feasible that persisting soluble phenolics could continue to react with enzymes [47], this equation offers a minimum estimate of %Phenolic Inhibition.

In bog incubations, phenolics significantly (p<0.001) inhibited $C_{tot}$, $CO_2$, and $CH_4$ production by 61 ± 12, 62 ± 16, and 54 ± 19%, respectively (Table 3; standard error calculating by propagating error from $Prod_0$ and $Prod_{sat}$). In incubated fen, phenolics significantly (p<0.001)

**Table 2. Michaelis-Menten parameters for $CO_2$, $CH_4$, and $C_{tot}$ (Fig 2: Panels a-d and e-g).**

| Habitat | Gas | $k_m$ | $v_m$ | $Prod_0$[a] | $Prod_{sat}$[b] | $R^2$ |
|---|---|---|---|---|---|---|
| Bog | $CO_2$ | 0.008 ± 0.004 | 0.08 ± 0.01 | 0.046 ± 0.004 | 0.13 ± 0.01 | 0.829 |
| | $CH_4$ | 0.003 ± 0.002 | 0.02 ± 0.003 | 0.014 ± 0.003 | 0.037 ± 0.004 | 0.617 |
| | $C_{tot}$ | 0.008 ± 0.003 | 0.11 ± 0.01 | 0.06 ± 0.006 | 0.17 ± 0.01 | 0.835 |
| Fen | $CO_2$ | 0.001 ± 0.002 | 0.17 ± 0.03 | 1.01 ± 0.02 | 1.18 ± 0.04 | 0.884 |
| | $CH_4$ | N/A[c] | N/A[c] | N/A[c] | N/A[c] | N/A[c] |
| | $C_{tot}$ | 0.001 ± 0.001 | 0.15 ± 0.04 | 1.38 ± 0.03 | 1.53 ± 0.05 | 0.808 |

Michaelis-Menten equations relate PVP concentration ($g \times mL^{-1}$) to $CO_2$, $CH_4$, and $C_{tot}$ production rate (in $\mu moles \times g^{-1} \times d^{-1}$). $k_m$ and $v_{max}$ were calculated using the Python-based SciPy library with the Optimize package and the curve_fit function. The standard deviation for these constants was determined via non-parametric bootstrapping (1,000 simulations). $R^2$ values reference the fit between modeled and measured values.

[a]$Prod_0$ ($\mu moles \times g^{-1} \times d^{-1}$) refers to the average control production rate (where PVP = 0 g $\cdot mL^{-1}$) and is equivalent to the y-intercept for the Michaelis-Menten curves.

[b]$Prod_{sat}$ ($v_{max}+Prod_0$) is an estimation of the production rate in PVP-saturated peat.

[c]Michaelis Menten parameters could not be determined for the relationship between PVP concentration and fen $CH_4$ production, as no significant relationship between these factors was identified.

inhibited $C_{tot}$ and $CO_2$ production, though to a lesser extent—10 ± 4 and 14 ± 4%, respectively. No apparent inhibition of $CH_4$ production in fen incubations was observed, as PVP concentration had no significant impact on $CH_4$ production rates.

## Pore-water analysis from field samples

Soluble phenolic content, presented in Table 4, was significantly influenced by habitat, depth, and site ($p < 0.001$). The ordinary least-squares regression model obtained using habitat and depth accounted for 78.6% of all observed variance in soluble phenolic content. Habitat had the strongest influence on soluble phenolic content (t = 9.83, coefficient = 4.77) with bog samples exhibiting 2.9 to >15 x higher soluble phenolic content than fen samples (Table 4). Depth had the next strongest influence (t = 3.76, coefficient = 0.07) with greater depths exhibiting generally higher soluble phenolic content. Site had the smallest impact (t = 2.39, coefficient = 1.10), with slightly higher phenolic content observed at site 2 (for both habitats).

## Discussion

We hypothesized that increasing PVP concentration would yield increasing $GHG_c$ production rates and that this relationship would follow an amended Michaelis-Menten (saturation) function (Fig 1). We hypothesized that this relationship would be more pronounced in bog vs. fen

**Table 3. Percent phenolic inhibition of $C_{tot}$, $CO_2$, and $CH_4$ production rates.**

| Gas Analyzed | Minimum %Phenolic Inhibition[a] | |
|---|---|---|
| | **Bog** | **Fen** |
| $C_{tot}$[b] | 61 ± 12 | 10 ± 4 |
| $CO_2$ | 62 ± 16 | 14 ± 4 |
| $CH_4$ | 54 ± 19 | N/A[c] |

[a]%Phenolic Inhibition estimates were calculated via Eq12.

[b]$C_{tot}$ represents the sum of $CO_2$ and $CH_4$ production rates.

[c]As PVP concentration had no significant impact on $CH_4$ production in fen peat, %Inhibition could not be determined.

**Table 4. Soluble phenolic content by site, depth, and habitat.**

| Site Number[a] | Depth Range (cm) | Soluble Phenolics ($\mu$M)[b] | |
|---|---|---|---|
| | | Bog | Fen |
| 1 | 1–5 | n/a[c] | < 2.5[d] |
| | 10–14 | 39 ± 0 | < 2.5[d] |
| | 20–24 | 31 ± 16 | 10 ± 5 |
| | 30–34 | 52 ± 7 | n/a[c] |
| | 40–44 | 52 ± 1 | n/a[c] |
| 2 | 1–5 | n/a[c] | 5 ± 5 |
| | 10–14 | 44 ± 15 | 13 ± 9 |
| | 20–24 | 45 ± 17 | 13 ± 11 |
| | 30–34 | 62 ± 13 | 14 ± 14 |
| | 40–44 | 53 ± 24 | 18 ± 14 |

[a]Coordinates for site 1 are 19.04758˚N, 68.35330˚W and 19.04658˚N, 68.35337˚W for bog and fen, respectively. Coordinates for site 2 are 19.04923˚N, 68.35559˚W and 19.04620˚N, 68.35443˚W for bog and fen, respectively.
[b]Soluble phenolics are calculated in gallic acid equivalents.
[c]Samples could not be collected at this depth/site/habitat combination.
[d]Indicates values below detection limit.

incubations, as prior pore-water analysis of our study site (Stordalen Mire) has revealed greater prevalence of compounds with high O/C, molecular weight, and aromaticity in the bog (indicative of higher phenolic content; [22]).

## Evaluating hypotheses

The correlation between PVP concentration and bog $GHG_c$ production was positive and robustly approximated using Michaelis-Menten curves. This held true for $CO_2$ and $C_{tot}$ production from fen peat. These findings are consistent with our hypothesis and can be reasonably attributed to PVP-phenolic interactions, which alleviate phenolic inhibition of microbial processes [27, 28, 46, 47]. Counter to our first hypothesis, we observed no significant relationship between PVP and fen $CH_4$ production, which, combined with the significant $CO_2$ responsivity, resulted in a significant positive correlation between PVP and $CO_2:CH_4$ production ratios. This observation suggests that soluble phenolics did not significantly inhibit $CH_4$ production in these incubations.

Also consistent with our hypothesis, we observed significantly higher soluble phenolic content and %phenolic inhibition of $GHG_c$ production in bog vs. fen peat. The difference in phenolic inhibition among habitats was particularly striking for methane production, given the lack of responsivity to PVP addition in the fen.

## Exploring habitat differences

The generally higher %Phenolic Inhibition observed in bog peat can likely be attributed to its higher soluble phenolic abundance, which would have allowed for more enzyme-phenolic interactions to occur.

There are three possible explanations for the unique lack of methane responsivity to phenolic removal in the fen: (1) methanogenic enzymes in the fen could have a limited affinity to bind with soluble phenolics; (2) bog peat could contain specific phenolic compounds that are particularly effective at binding to methanogenic enzymes; (3) a certain threshold for soluble

phenolic content could be necessary for interactions between methanogenic enzymes and phenolics to occur, and our fen incubations were below this threshold.

Methanogen community structure differs significantly among peatland habitats [53–55, 57]. One could argue that this observation supports explanation 1 (that methanogenic enzymes in the fen could have a limited affinity to bind with soluble phenolics)—as it is feasible that differing methanogenic community structures lead to differing enzyme speciation. However, significantly higher diversity among the methanogen community within fens has been observed relative to bogs [53–55], rendering the possibility that fens do not possess any significant pool of methanogenic enzymes capable of binding to soluble phenolics unlikely.

A diverse pool of soluble phenolics have been detected in bioreactors containing living *Sphagnum spp*. [58, 59]. While this implicates explanation 2 (that bog peat could contain specific phenolic compounds that are particularly effective at binding to methanogenic enzymes) as feasible, a more detailed analysis of soluble phenolic speciation across peatland permafrost thaw gradients is necessary to fully assess the impact of phenolic speciation on the regulation of peat decay. Further research, potentially involving phenolic addition to fen, is also necessary for investigating possibility 3 (that a certain threshold for soluble phenolic content could be necessary for interactions between methanogenic enzymes and phenolics to occur).

## Incubation-field comparison

Incubations are imperfect representations of *in situ* phenomena, as perturbations imposed by their sampling, setup, and implementation are inevitable. Thorough analyses—conducted using field and incubation data taken from our study site—have thus been undertaken as a means of quantifying these perturbations [55, 60]. Findings demonstrate that the incubations methods employed herein yield C mineralization rates and $CO_2$:$CH_4$ production ratios that are largely consistent with *in situ* observations [60]. They also indicate that, with respect to microbial communities and carbon cycle geochemistry, these incubation largely reproduce field results, indicating functional consistency [55].

Crucial to this consistency is the implementation of a 25-day pre-incubation period (which was incorporated in this study). This period has been shown to significantly lessen perturbations imposed upon the microbial community immediately following setup [55]. This observation, combined with the consistency in greenhouse gas production rates between the field and incubations (observed after pre-incubation) [60], indicates that any enzymes degraded during freezer storage (-20˚C; [61]) are likely replenished during this pre-incubation period. Further supporting this supposition is the observation that $CO_2$ and $CH_4$ are produced rapidly following permafrost thaw in Finnish permafrost peat [62]—as this could not be accomplished without enzymatic rebound.

## Geographic variation

Different locations of the same habitat (bog or fen) within Stordalen Mire exhibited significant differences in soluble phenolic content, indicating that the extent of inhibition imposed by soluble phenolics likely differs with landscape heterogeneity. To ascertain the extent of variability in phenolic regulation within and between habitats, it is necessary to broaden assessments to numerous sites within the Mire. Such assessments would ideally incorporate both soluble phenolic content measurements and % Phenolic Inhibition estimates.

To capture the effects of soluble phenolics on wider geographic scales, it is necessary to disentangle the relative impact of biotic (vegetation, humification indices, microbial community dynamics) and abiotic (temperature, pH) factors upon enzyme-phenol interactions. To do so, a combination of field-based (incorporating samples from multiple sites, with numerous

replicate cores, at differing thaw progression stages) and incubation-based assessments is necessary.

When considering the role of vegetation, it is recommended that plant species distribution be considered. This recommendation is based on the observation that exogenous concentrations of several phenolic compounds (including *trans*-Sphagnum acid, *cis*-Sphagnum acid, *trans*-sphagnum acid ethyl ester, hydroxybutenolide, *p*-hydroxybenzoic acid, *p*-coumaric acid, and *trans*-cinnamic acid) were consistently (3–14.5 x) higher in bioreactors containing *S. cuspidatum* than *S. fallax* [63]. As this suggests significant variation in the affinity for different *Sphagnum* species to release certain soluble phenolics, it is possible the abundance of hydric species at our study site—e.g. *S. balticum* [49]—could have significantly influenced the extent of phenolic regulation observed.

## Climate implications

If applicable to wide geographic scales, our results indicate that soluble phenolics could contribute to two characteristic attributes of peat bogs: (1) their extraordinary recalcitrance [20, 21]—evidenced by their greater responsivity to soluble phenolic removal—and (2) their generally elevated $CO_2:CH_4$ ratios—evidenced by the significant phenolic regulation of methanogenesis in bog but not fen peat.

These effects have important implications for the climate. Numerous studies indicate that the introduction of molecular oxygen can stimulate phenol oxidase, which degrades phenolics, causing enhanced decay of organic matter [9, 24, 25, 64]. This effect is thought to be exacerbated by (1) the onset of oxic respiration and (2) increased diversity and abundance of bacterial species capable of catabolizing phenolics—both of which have been observed following drainage of *Sphagnum*-rich peat [65]. These findings are the foundation of an assertion that rapid peat decay could occur following drainage (via anthropogenic disturbances) or exposure to drought (from rising temperatures) [9, 24, 25].

The positive correlation we observed between depth and soluble phenolic abundance is consistent with oxygen infiltration significantly lowering phenolic content, as oxygen infiltration occurs more regularly at shallow depths [11]. However, inconsequential effects of oxygen introduction on phenolic abundance have been observed in some peatland sites [40, 66–68]. It has been suggested that peat bogs are particularly resilient to the effects of drainage, given that comparatively lower microbial community shifts following long-term drainage have been observed in bogs relative to fens [68]. Further research on this subject is needed before we can assess the implications of our phenolic inhibition estimates on C mineralization shifts following water-table fluctuations.

Furthermore, the significant differences in phenolic abundance observed between bog and fen peat indicate that habitat transitions due to thaw could significantly impact soluble phenolic content. Satellite studies have revealed widespread expansion of northern peatland shrubs over multidecadal timeframes [48, 69], while warming experiments have induced significant loss of *Sphagnum* [70]. If the stronger influence of phenolics on C mineralization observed in bogs applies to wide geographic scales, the cumulative inhibition of peatland C mineralization by phenolics could be significantly offset by habitat transition.

## Conclusions

We employed anaerobic incubation experiments of bog and fen peat (collected from Stordalen Mire, Sweden) inoculated with varying concentrations of PVP—a compound known to decrease phenolic effectiveness—as a means of investigating the impact of phenolics on C mineralization. We estimated that the extent to which phenolics inhibit $CO_2$ production is

significantly higher in the bog—62 ± 16%—than the fen—14 ± 4%. This difference was found to be more substantial with regards to methane production—wherein phenolic inhibition for the bog was estimated at 54 ± 19%, while the fen demonstrated no apparent inhibition. Soluble *in situ* phenolic content was significant higher (2.9 to >15 x) in bog vs. fen samples, indicating that the difference in inhibitory effects could be due, in part, to soluble phenolic availability. If applicable across different peatland sites, these findings suggest that soluble phenolics could contribute to peat bogs' extraordinary recalcitrance [11, 20, 21], and generally high $CO_2$:$CH_4$ ratios [16, 18].

## Supporting information

**S1 File. Incubated bog and fen production rates for $CH_4$, $CO_2$, and $C_{tot}$ ($CO_2$+$CH_4$).** Two experiments were conducted, using bog and fen peat (column ID = "Habitat"). Each experiment had 18 samples (column ID = "Samp. #"), each with one of six possible PVP concentrations, ranging from 0–0.064 g/mL (column ID = "PVP Conc."). Slope and $R^2$ values reference linear regression fits for gas production timeseries (Day vs. $CO_2$, $CH_4$, or $C_{tot}$ production), with final production rates in μmoles × g dry $wt^{-1}$ × $d^{-1}$). Outliers were determined when the production rates of any of $CO_2$, $CH_4$, or $C_{tot}$ differed from modeled values by >2 standard deviations (Methods, Statistical Analysis).
(CSV)

## Acknowledgments

We want to thank collaborators from the department of Earth, Ocean, and Atmospheric Sciences at The Florida State University (FSU) for their assistance with (1) UV-vis spectrometry —namely Dr. Sven Kranz—and data analysis—namely Drs. Thomas Kelly and Michael Stukel. In addition, we gratefully acknowledge the staff of Abisko Naturvetenskapliga, and the 2018 IsoGenie field sampling team for collecting material used for the incubations.

We also want to thank the Isogenie Project Coordinators—including Scott R. Saleska[1]*, Virginia I. Rich[2], Patrick M. Crill[3], Jeffrey P Chanton[4], Gene W. Tyson[5], Ruth K. Varner[6], Matthew B. Sullivan[2], Steven Frolking[6], Changsheng Li[6], Eoin L. Brodie[7], William J. Riley[7]— and the EMERGE 2021 Field Team—including Ruth K. Varner6**, Sophie A. Burke[6], Apryl L. Perry[6], Jessica M. Szetela[1], Scott R, Saleska[1], Helene Saleska[1], Carmody K. McCalley[3], and Patrick M. Crill[3].

1 Department of Ecology and Evolutionary Biology, University of Arizona, Tucson, AZ, United States of America

2 Department of Microbiology, The Ohio State University, Columbus, OH, United States of America

3 Department of Geological Sciences, Stockholm University, Stockholm, Sweden

4 Department of Earth, Ocean, and Atmospheric Science, Florida State University, Tallahassee, FL, United States of America

5 School of Earth & Environmental Science, Queensland University of Technology, Brisbane, Australia

6 Department of Earth Sciences, University of New Hampshire, Durham, NH, United States of America

7 Climate and Ecosystem Sciences Division, Lawrence Berkeley National Laboratory, Berkeley, CA, United States of America

* Corresponding and lead member of the Isogenie Project Coordinators
E-mail: saleska@arizona.edu

** Corresponding and lead member of the EMERGE 2021 Field Team
E-mail: ruth.varner@unh.edu

## Author Contributions

**Conceptualization:** Alexandra B. Cory, Jeffrey P. Chanton, Robert G. M. Spencer.

**Data curation:** Alexandra B. Cory, Olivia C. Ogles.

**Formal analysis:** Alexandra B. Cory, Jeffrey P. Chanton, Rachel M. Wilson.

**Funding acquisition:** Jeffrey P. Chanton, Virginia I. Rich, Rachel M. Wilson.

**Investigation:** Alexandra B. Cory, Jeffrey P. Chanton, Olivia C. Ogles.

**Methodology:** Alexandra B. Cory, Jeffrey P. Chanton, Carmody K. McCalley.

**Project administration:** Jeffrey P. Chanton.

**Resources:** Jeffrey P. Chanton, Carmody K. McCalley.

**Software:** Alexandra B. Cory.

**Supervision:** Jeffrey P. Chanton, Rachel M. Wilson.

**Validation:** Alexandra B. Cory.

**Visualization:** Alexandra B. Cory.

**Writing – original draft:** Alexandra B. Cory.

**Writing – review & editing:** Alexandra B. Cory, Jeffrey P. Chanton, Robert G. M. Spencer, Virginia I. Rich, Rachel M. Wilson.

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
