## [Decision Letter · Decision Letter 0]

13 Jul 2021

PONE-D-21-16549

Quantifying the Inhibitory Impact of Soluble Phenolics on Carbon Mineralization from Sphagnum-rich Peatlands

PLOS ONE

Dear Dr. Cory,

Thank you for submitting your manuscript to PLOS ONE. After careful consideration, we feel that it has merit but does not fully meet PLOS ONE’s publication criteria as it currently stands. Therefore, we invite you to submit a revised version of the manuscript that addresses the points raised during the review process.

We look forward to receiving your revised manuscript.

Kind regards,

Muhammad Riaz

Academic Editor

PLOS ONE

“Funding for this study was provided by the Genomic Science Program of the United States Department of Energy Office of Biological and Environmental Research Grants (DE-SC0010580 & DE-SC0016440). Additional funding was provided by the EMERGE Biology Integration Institute of the National Science Foundation (NSF Award # 2022070).”

 “Grant 1

Recipient: V.I.R.

Grant number: DE-SC0010580

Funding Source: the Genomic Science Program of the United States Department of Energy Office of Biological and Environmental Research Grants

URL: https://genomicscience.energy.gov/

Grant 2

Recipient: V.I.R.

Grant number: DE-SC0016440

Funding Source: the Genomic Science Program of the United States Department of Energy Office of Biological and Environmental Research Grants

URL: https://genomicscience.energy.gov/

Grant 3

Recipient: V.I.R.

Grant Number: 2022070

Funding Source: the EMERGE Biology Integration Institute of the National Science Foundation

URL: https://www.nsf.gov/funding/pgm_summ.jsp?pims_id=505684

3. One of the noted authors is a group or consortium “IsoGenie Project Coordinators.” In addition to naming the author group, please list the individual authors and affiliations within this group in the acknowledgments section of your manuscript. Please also indicate clearly a lead author for this group along with a contact email address.

Additional Editor Comments (if provided):

I have received three review reports for your manuscript. The reviewers are of view that your manuscript is interesting but needs to be revised to address various concerns. I invite you to revise the manuscript by giving due consideration to the comments and suggestions of reviewers particularly reviewer 2 who has raised some very critical issues about your manuscript

Reviewers' comments:

Reviewer's Responses to Questions

**Comments to the Author**

1. Is the manuscript technically sound, and do the data support the conclusions?

Reviewer #1: Yes

Reviewer #2: Partly

Reviewer #3: Yes

2. Has the statistical analysis been performed appropriately and rigorously? 

Reviewer #1: Yes

Reviewer #2: Yes

Reviewer #3: Yes

3. Have the authors made all data underlying the findings in their manuscript fully available?

Reviewer #1: Yes

Reviewer #2: Yes

Reviewer #3: Yes

4. Is the manuscript presented in an intelligible fashion and written in standard English?

Reviewer #1: Yes

Reviewer #2: Yes

Reviewer #3: Yes

5. Review Comments to the Author

Reviewer #1: Soluble phenolics have been invoked as potentially significant inhibitors of bog decomposition due to their propensity to suppress microbial metabolism and inhibit cell growth. Though the inhibitory effects of soluble phenolics are generally accepted, the extent to which they inhibit C mineralization in Sphagnum peatlands remains unclear. The study clarify the impacts of soluble phenolics on bog C mineralization in three stages of bog decomposition, and found that soluble phenolics inhibit, at minimum, 57 ± 16% of total C (CO2+CH4) mineralization. These findings indicated that soluble phenolics play a significant role in regulating bog peat stability in the face of decomposition. I believe the results is considerable meaningufl. However, there are some litter problens needed to be solved. First, the relationship between polyvinylpyrrolidone (PVP) concentration vs. soluble phenolics shouled be showed in your measured data. Second, each treatment only with three replicates, How do you handle outliers?

Reviewer #2: The paper is generally well written and well structured, and the subject of inhibitory effects in sphagnum peat is interesting and relevant in an environmental context. However, the ms contains several flaws the reason for which I do not recommend publication in PlosOne.

The discussion seems like a recommendation for further research. The paper contains an extensive Introduction, followed by very few results, and a Discussion which is in fact almost completely a recommendation for further research. To me the paper reads like an extensive M&M Section. This is also reflected by the fact that citations 1-44, out of 56 in total, are all included already in the Introduction (and the References, Acknowledgements and title page are 25% of the text, 2000/6000 words).

A major problem is that the Methodology is humble. I suggest to the authors to follow their own recommendations and add more data to validate the results in an environmental context:

The authors name three main processes: hydrolysis, fermentation and methanogenesis (L64-65) and state that these are all stages of decomposition (L81-82). Though, how strong are the results of phenolics inhibiting peat OM decay in water-saturated conditions relative to aerobic decomposition? Isn’t this a fraction of it? In “These findings are consistent with other studies that have indicated that soluble phenolics play a significant role in regulating bog peat stability in the face of decomposition” (L36-38) it is not indicated that it all refers to anaerobic conditions, but “decomposition” is often understood as aerobic decomposition in peatlands. In L 99-100 it is mentioned that peat typically resides in anaerobic conditions, being the reason for the authors to study the effects under anaerobic conditions, though most of the mass loss occurs, of course, in the surface under aerobic conditions. Adding a reference of GHGc production for aerobic conditions (without PVP) is required to give the results studied here under anaerobic conditions relevance in an environmental context.

Or some other “context results” such as is mentioned in L323-327, why is this beyond the scope of this study? Like you say in L328-331 this variation is existent and relevant. Why, for example, no replicate core was taken? Or different depths? It is now a single sample of a single depth of a single mire. This is a good starting point for the model, but not sufficient material for a paper.

In the M&M information is missing on

- the water table at the core location. Is 9-19 cm (L146) in the catotelm?

- The reasoning for the choice of this depth interval

- The size of the total peat sample

- Incubation temperature

Minor points:

L145 check spelling of the corer name

L149 and 154. 6*3=21?

the ms contains unnecessary inaccuracies:

- Abbreviations should be checked

- Reference list should be revised, it contains many inconsistencies, for example the points, spaces and comma’s in authors names, capitals, letter size, italic, etc

Reviewer #3: In general, phenolic-enzyme interaction is essential to understand peatland decomposition. This manuscript (MS) is well in line with understanding peatland biogeochemical response towards carbon mineralization. The current MS has some errors that need to be corrected before further processing. A main concern is related to Methods section. Overall, a nice simple experimentation reported in a clear and concise manner that will contribute to advance our understanding of how peatland ecosystems succeed to accumulate C through geological times.

Abstract

Line 24-26 It is well evident in literature that peatlands have a slow decomposition rate due to low pH, anoxic conditions etc. However, little is known about phenolic-enzyme interaction with contradictory results. So, I suggest to re-write the following statement “The mechanisms controlling the extraordinarily slow carbon (C) mineralization rates characteristic of Sphagnum-rich peatlands (“bogs”) remain somewhat elusive, despite decades of research on this topic.”

Introduction

Line 44-45 It is irrelevant in comparison to Line 45-48 “Over the last ~150 years, the peatland 46 carbon sink has diminished with estimates of present climatic impacts ranging from slightly cooling (-0.7 W x m-2; instantaneous box-model estimate) to slightly warming (+0.6 Pg CO2- equiv y–1; field flux estimate)”. Therefore, remove the following lines “For the past 8000-11,000 years, peatland C deposition has had a net cooling effect on the climate.”

Line 58-59 As suggested in the abstract section, it is a general statement and in literature, it is evident that decomposition in peatlands is slow. So, try to be specific such as cause of this remains elusive particularly in phenolic-enzyme interaction.

Methods

Line 144-146 It is not clear that how peat sampling was carried out in the study site such as random peat samples, transect line etc. It is suggested to add more details.

Line 146-147 The main idea of current MS deals with phenolic-enzyme interactions and the response of polyvinylpyrrolidone (PVP) to CO2 and CH4 production. Peat samples were stored at -20 ℃ and the total duration of storage is not mentioned. At such low temperature, enzymes are denatured, and their activities are not stable (see reference Dunn, C., Jones, T. G., Girard, A., and Freeman, C. (2014). Methodologies for extracellular enzyme assays from wetland soils. Wetlands 34, 9–17).

The author lacks a conceptual understanding of the enzyme activities in response to temperature sensitivity. Therefore, it is recommended to do a short trial with fresh peat samples to justify the presented results.

Line 149 Total treatments six with three replicates make 18 vials. In MS, 21 vials are mentioned, which is a mistake and if not then clarify it.

6. PLOS authors have the option to publish the peer review history of their article (what does this mean?). If published, this will include your full peer review and any attached files.

Reviewer #1: No

Reviewer #2: No

Reviewer #3: No

---

## [Author Response · Author response to Decision Letter 0]

2 Nov 2021

Reviewer: 1

Comment #1: Soluble phenolics have been invoked as potentially significant inhibitors of bog decomposition due to their propensity to suppress microbial metabolism and inhibit cell growth. Though the inhibitory effects of soluble phenolics are generally accepted, the extent to which they inhibit C mineralization in Sphagnum peatlands remains unclear. The study clarify the impacts of soluble phenolics on bog C mineralization in three stages of bog decomposition, and found that soluble phenolics inhibit, at minimum, 57 ± 16% of total C (CO2+CH4) mineralization. These findings indicated that soluble phenolics play a significant role in regulating bog peat stability in the face of decomposition. I believe the results is considerable meaningufl. However, there are some litter problens needed to be solved. First, the relationship between polyvinylpyrrolidone (PVP) concentration vs. soluble phenolics shouled be showed in your measured data. 

Response: 

To address this request, we implemented two published methods (Folin-Ciolcateu and Somers) for determining the total phenolic of our incubated pore-water. Unfortunately, with each method the presence of PVP interfered significantly with the absorbance measured via UV-vis spectroscopy. As such, we could not directly fulfill this request. 

We were, however, able to indirectly analyze the relationship between soluble phenolic content and PVP by expanding our study to a second environment at Stordalen mire, a fen ecosystem within 50 meters of our bog site. We measured soluble phenolic content at two fen and two bog sites and found that the fen had significantly lower soluble phenolic content. We ran a second set of incubations of fen material and found that the PVP had a significantly lower (though still measurable) impact on anaerobic decomposition in fen incubations, consistent with the fen having lower phenolic content.

Comment #2: Second, each treatment only with three replicates, How do you handle outliers?

Response: In the fen, we had one outlier, which was the measured GHGc production rate of one vial at PVP= 0.004 g/mL. This datapoint was initially identified as a potential outlier when we calculated Michaelis Menten (M.M) parameters associated with all measured data (before outlier exclusion). The M.M. parameters obtained from this initial model did not converge with measured values. After excluding the potential outlier from the M.M. parameter determination, we plotted measured vs. modeled values (which did converge) and calculated the squared residuals for each measured value. With a squared residual value that was greater than 2 standard deviations from the modeled value, we determined that this value was in fact an outlier.

We also had one outlier in the bog (which also represented the measured GHGc production rate in one vial, where PVP=0.064 g/mL). This outlier was also > 2 standard deviations from the modeled value.

After excluding outliers, we re-ran the M.M. model. We retained all original data in our average and standard deviation calculations for measured values. As such, all measured data is captured in Fig. 1. This information has been added to the Methods section (lines 327-330) and Results (Lines 358-359 and 382-383). 

Reviewer: 2

Comment #1: The paper is generally well written and well structured, and the subject of inhibitory effects in sphagnum peat is interesting and relevant in an environmental context. However, the ms contains several flaws the reason for which I do not recommend publication in PlosOne.

The discussion seems like a recommendation for further research. The paper contains an extensive Introduction, followed by very few results, and a Discussion which is in fact almost completely a recommendation for further research. To me the paper reads like an extensive M&M Section. This is also reflected by the fact that citations 1-44, out of 56 in total, are all included already in the Introduction (and the References, Acknowledgements and title page are 25% of the text, 2000/6000 words). 

Response: We expanded the discussion to 5+ pages. It now incorporates an assessment of the differences between bog and fen habitats with respect to %Phenolic Inhibition and soluble phenolic content (Lines 460-484). We also included a section on incubation-field comparisons (Lines 485-50) and geographic variation (Lines 502-522).

Comment #2: A major problem is that the Methodology is humble. I suggest to the authors to follow their own recommendations and add more data to validate the results in an evironmental context: The authors name three main processes: hydrolysis, fermentation and methanogenesis (L64-65) and state that these are all stages of decomposition (L81-82). Though, how strong are the results of phenolics inhibiting peat OM decay in water-saturated conditions relative to aerobic decomposition? Isn’t this a fraction of it? 

Response: Most peatlands are net C sinks but CH4 sources (Ussiri and Lal, 2017; Gorham, 1992; Frolking et al., 2011). Given that CH4 is a significantly stronger greenhouse gas than CO2 (45x on a 100-year timescale; Neubauer and Megonigal, 2015), the climatic impacts of peatlands depend heavily on their CO2:CH4 production ratios. Given that the vast majority of CH4 is produced anaerobically, the impacts of soluble phenolic on OM decay in water-saturated conditions has important implications for the climate. A discussion of the climatic relevance of anaerobic decomposition is included in Lines 61-71.

Comment #3: In “These findings are consistent with other studies that have indicated that soluble phenolics play a significant role in regulating bog peat stability in the face of decomposition” (L36-38) it is not indicated that it all refers to anaerobic conditions, but “decomposition” is often understood as aerobic decomposition in peatlands. 

Response: We clarified that this study focuses on anaerobic decomposition in the abstract (Lines 30, 32, and 34; The content of the original sentence from L36-38 was altered in order to incorporate an assessment of bog vs. fen observations. The replacement sentence did not require the alteration that you suggest.) This was also clarified in the Introduction (Lines 69-71).

Comment #4: In L 99-100 it is mentioned that peat typically resides in anaerobic conditions, being the reason for the authors to study the effects under anaerobic conditions, though most of the mass loss occurs, of course, in the surface under aerobic conditions. Adding a reference of GHGc production for aerobic conditions (without PVP) is required to give the results studied here under anaerobic conditions relevance in an environmental context.

Response: A cross-analysis between aerobic and anaerobic CO2 and CH4 production was undertaken by Treat et al. (2014) using Alaskan peat. Their analysis indicates that anaerobic production of CO2 is approximately half that of aerobic production, while anaerobic production of CH4 was over twice that of aerobic incubations. Given the relatively high radiative forcing of methane (45 times CO2 on a 100-year timescale; Neubauer and Megonigal, 2015), the net radiative forcing of the anaerobic incubations from Treat et al. (2014) was higher than that of aerobic incubations. 

Greenhouse gas flux measurements from our study site, Stordalen Mire, indicate that the anaerobic realm has a similarly impactful role here. This is evident from the significant flux of CH4 to the atmosphere (89 gC/m2; Backstrand et al., 2010), which is formed primarily in the anaerobic realm. CO2 has a negative flux to the atmosphere (-20 gC/m2; Backstrand et al., 2010) which indicates that rates of aerobic C mineralization (which primarily produces CO2) are not currently high enough to yield a warming impact.

A discussion of the climatic relevance of anaerobic decomposition has been included in Lines 61-71.

Comment #5: Or some other “context results” such as is mentioned in L323-327, why is this beyond the scope of this study? Like you say in L328-331 this variation is existent and relevant. Why, for example, no replicate core was taken? Or different depths? It is now a single sample of a single depth of a single mire. This is a good starting point for the model, but not sufficient material for a paper.

Response: We employed a two-fold approach to add context to our original findings. 

First, we repeated this experiment on fen peat. In this additional experiment, we used peat collected from the same depth and date as the original bog material. The fen site was located within 50 m of the bog site. The incubation protocol was kept consistent.

Second, we examined the relationship between habitat and depth vs. soluble phenolic content. Specifically, we examined differences between bog and fen habitats. For each of these habitats, we measured soluble phenolic content at two sites. For each site, we obtained three depth profiles, each of which contained up to four depth intervals (depending on substrate availability), spanning a 43-cm range (unless substrate availability precluded collection). 

Respective amendments to the manuscript are included in Lines 203; 227; 261-271; 272-279; 380-395; 408-436; 446-522; Fig. 2; and Tables 2-3. 

Comment #6: In the M&M information is missing on

- the water table at the core location. Is 9-19 cm (L146) in the catotelm? -The reasoning for the choice of this depth interval

Response: Included the following explanation (Lines 188-190): “This depth range was selected because this depth peat is generally below the water table, facilitating anaerobic decomposition (which is the focus of this study).”

Comment #7: - The size of the total peat sample

Response: We included the length and diameter of the bog and fen cores along with the weights of the 9-19 cm subsections (which were used for incubation analysis; Lines 186-191).

Comment #8: - Incubation temperature

Response: This was included in the original manuscript, and is retained in the revised version (Lines 220-221): “We stored the vials in total darkness at room temperature (20-22°C)”

Comment #9: Minor points:

L145 check spelling of the corer name

Response: Changed “Eijenkamp” to “Eijelkamp”. 

Comment #10: L149 and 154. 6*3=21?

Response: Amended 21 to 18. 

Comment #11: the ms contains unnecessary inaccuracies:

- Abbreviations should be checked

Response: Quotation marks included around the first occurrence “Prod0”, “GHGc”, and “PVP”.

Removed abbreviation for Michaelis-Menten (M.M.) in the abstract, as it is not abbreviated elsewhere in the manuscript.

Comment #12: Reference list should be revised, it contains many inconsistencies, for example the points, spaces and comma’s in authors names, capitals, letter size, italic, etc.

Response: Reference list has been revised. 

References for Reviewer 2:

Bäckstrand K, Crill PM, Jackowicz-Korczynski M, Mastepanov M, Christensen TR, Bastviken D. Annual carbon gas budget for a subarctic peatland, Northern Sweden. Biogeosciences. 2010 Jan 11;7(1):95-108: 

Frolking, S., Talbot, J., Jones, M. C., Treat, C. C., Kauffman, J. B., Tuittila, E. S., & Roulet, N. (2011). Peatlands in the Earth’s 21st century climate system. Environmental Reviews, 19(NA), 371-396

Gorham E. Northern peatlands: role in the carbon cycle and probable responses to climatic warming. Ecological applications. 1991 May;1(2):182-95

Neubauer SC, Megonigal JP. Moving beyond global warming potentials to quantify the climatic role of ecosystems. Ecosystems. 2015 Sep;18(6):1000-13.

Ussiri D, Lal R. The Global Carbon Inventory. Carbon Sequestration for Climate Change Mitigation and Adaptation. 2017 (pp. 77-102). Springer, Cham.

Reviewer: 3

Comment #1: In general, phenolic-enzyme interaction is essential to understand peatland decomposition. This manuscript (MS) is well in line with understanding peatland biogeochemical response towards carbon mineralization. The current MS has some errors that need to be corrected before further processing. A main concern is related to Methods section. Overall, a nice simple experimentation reported in a clear and concise manner that will contribute to advance our understanding of how peatland ecosystems succeed to accumulate C through geological times.

Abstract

Line 24-26 It is well evident in literature that peatlands have a slow decomposition rate due to low pH, anoxic conditions etc. However, little is known about phenolic-enzyme interaction with contradictory results. 

So, I suggest to re-write the following statement “The mechanisms controlling the extraordinarily slow carbon (C) mineralization rates characteristic of Sphagnum-rich peatlands (“bogs”) remain somewhat elusive, despite decades of research on this topic.”

Response: Changed to “The mechanisms controlling the extraordinarily slow carbon (C) mineralization rates characteristic of Sphagnum-rich peatlands (“bogs”) are not fully understood, despite decades of research on this topic.” (Lines 23-25.)

Comment #2: Introduction

Line 44-45 It is irrelevant in comparison to Line 45-48 “Over the last ~150 years, the peatland 46 carbon sink has diminished with estimates of present climatic impacts ranging from slightly cooling (-0.7 W x m-2; instantaneous box-model estimate) to slightly warming (+0.6 Pg CO2- equiv y–1; field flux estimate)”. Therefore, remove the following lines “For the past 8000-11,000 years, peatland C deposition has had a net cooling effect on the climate.”

Response: Changed to: “While Peatland C deposition has had a significant cooling effect on the climate through much of the Holocene [7], this effect has diminished over the last ~150 years. Present climatic impacts range from slightly cooling (-0.7 W x m-2; instantaneous box-model estimate) [7] to slightly warming (+0.6 Pg CO2-equiv y–1; field flux estimate) [5].” (Lines 47-51.)

Comment #3: Line 58-59 As suggested in the abstract section, it is a general statement and in literature, it is evident that decomposition in peatlands is slow. So, try to be specific such as cause of this remains elusive particularly in phenolic-enzyme interaction.

Response: To clarify this point, we have re-arranged the introduction, and replaced the original sentence with Lines 77-80: “Certain abiotic factors partially explain the slower decomposition rates typically observed in bogs, such as lower pH (~4.5 in bogs, 7-8 in fens; Shotyk, 1988) and availability of TEAs. However, they do not fully account for observed greenhouse gas fluxes.” 

Comment #4: Methods

Line 144-146 It is not clear that how peat sampling was carried out in the study site such as random peat samples, transect line etc. It is suggested to add more details.

Response: An explanation has been added to Lines 185-186: “Sampling was random and attention was paid carefully not to sample in previously cored sites.”

Comment #5: Line 146-147 The main idea of current MS deals with phenolic-enzyme interactions and the response of polyvinylpyrrolidone (PVP) to CO2 and CH4 production. Peat samples were stored at -20 ℃ and the total duration of storage is not mentioned. At such low temperature, enzymes are denatured, and their activities are not stable (see reference Dunn, C., Jones, T. G., Girard, A., and Freeman, C. (2014). Methodologies for extracellular enzyme assays from wetland soils. Wetlands 34, 9–17).

The author lacks a conceptual understanding of the enzyme activities in response to temperature sensitivity. Therefore, it is recommended to do a short trial with fresh peat samples to justify the presented results.

Response: We included a section in the discussion aimed at addressing this issue, titled “Incubation-Field Comparison” (Lines 485-501). We also included the duration of freezer storage (20 and 29 months for bog and fen, respectively) in Line 192. 

We acknowledged that incubations are imperfect representations of in-situ phenomena, as perturbations imposed by their sampling, setup, and implementation are inevitable. For this reason, thorough analyses—conducted using field and incubation data taken from our study site—have previously been undertaken as a means of quantifying these perturbations (Hodgkins et al., 2015; Wilson et al., 2021). Findings demonstrate that the incubations methods employed herein yield C mineralization rates and CO2:CH4 production ratios that are largely consistent with in situ observations (Hodgkins et al., 2015). They also indicate that, with respect to microbial communities and carbon cycle geochemistry, these incubations largely reproduce field results, indicating functional consistency (Wilson et al., 2021). 

Crucial to this consistency is the implementation of a 25-day pre-incubation period (which was incorporated in this study). This period has been shown to significantly lessen the effects of perturbations imposed upon the microbial community immediately following setup (Wilson et al., 2021). This observation, combined with the consistency in greenhouse gas production rates observed pre-incubation (Hodgkins et al., 2015), indicates that replenishment of enzymes degraded during freezer storage (-20ºC; Dunn et al., 2014) likely occurs during this period. Further supporting this supposition is the observation that CO2 and CH4 are produced rapidly following permafrost in Finnish permafrost peat (Voigt et al., 2019)—as this could not be accomplished without enzymatic rebound. In addition, the soils used in this study freeze and thaw several times over the course of a year during winter periods.

Comment #6: Line 149 Total treatments six with three replicates make 18 vials. In MS, 21 vials are mentioned, which is a mistake and if not then clarify it.

Response: Changed from 21 to 18. 

References for Reviewer 3:

Hodgkins SB, Chanton JP, Langford LC, McCalley CK, Saleska SR, Rich VI, Crill PM, Cooper WT. Soil incubations reproduce field methane dynamics in a subarctic wetland. Biogeochemistry. 2015 Nov;126(1):241-9.

Voigt C, Marushchak ME, Mastepanov M, Lamprecht RE, Christensen TR, Dorodnikov M, Jackowicz‐Korczyński M, Lindgren A, Lohila A, Nykänen H, Oinonen M. Ecosystem carbon response of an Arctic peatland to simulated permafrost thaw. Global change biology. 2019 May;25(5):1746-64.

Wilson RM, Zayed AA, Crossen KB, Woodcroft B, Tfaily MM, Emerson J, Raab N, Hodgkins SB, Verbeke B, Tyson G, Crill P. Functional capacities of microbial communities to carry out large scale geochemical processes are maintained during ex situ anaerobic incubation. PloS one. 2021 Feb 25;16(2):e0245857.

---

## [Editor Report · Decision Letter 1]

3 Jan 2022

Quantifying the Inhibitory Impact of Soluble Phenolics on Anaerobic Carbon Mineralization in a Thawing Permafrost Peatland

PONE-D-21-16549R1

Dear Dr. Cory,

We’re pleased to inform you that your manuscript has been judged scientifically suitable for publication and will be formally accepted for publication once it meets all outstanding technical requirements.

Kind regards,

Muhammad Riaz

Academic Editor

PLOS ONE

Additional Editor Comments (optional):

After assessment of your revised manuscript, I am pleased to inform you that I have accepted your manuscript for publication.
---

## [Editor Report · Acceptance letter]

10 Jan 2022

PONE-D-21-16549R1 

Quantifying the Inhibitory Impact of Soluble Phenolics on Anaerobic Carbon Mineralization in a Thawing Permafrost Peatland 

Dear Dr. Cory:

I'm pleased to inform you that your manuscript has been deemed suitable for publication in PLOS ONE. Congratulations! Your manuscript is now with our production department. 

Kind regards, 

on behalf of

Dr. Muhammad Riaz 

Academic Editor

PLOS ONE